# Electrophysiological Markers of Visuospatial Attention Recovery after Mild Traumatic Brain Injury

**DOI:** 10.3390/brainsci9120343

**Published:** 2019-11-27

**Authors:** Julie Bolduc-Teasdale, Pierre Jolicoeur, Michelle McKerral

**Affiliations:** 1Centre for Interdisciplinary Research in Rehabilitation (CRIR), IURDPM, CIUSSS du Centre-Sud-de-l’Île-de-Montréal, Montreal, QC H3S 1M9, Canada; juliebolduct@me.com; 2Department of Psychology, Université de Montréal, Montreal, QC H3C 3J7, Canada; pierre.jolicoeur@umontreal.ca

**Keywords:** mTBI, event-related potentials, visual–attentional processing, brain connectivity, neuropsychological measures, postconcussion symptoms

## Abstract

Objective: Attentional problems are amongst the most commonly reported complaints following mild traumatic brain injury (mTBI), including difficulties orienting and disengaging attention, sustaining it over time, and dividing attentional resources across multiple simultaneous demands. The objective of this study was to track, using a single novel electrophysiological task, various components associated with the deployment of visuospatial selective attention. Methods: A paradigm was designed to evoke earlier visual evoked potentials (VEPs), as well as attention-related and visuocognitive ERPs. Data from 36 individuals with mTBI (19 subacute, 17 chronic) and 22 uninjured controls are presented. Postconcussion symptoms (PCS), anxiety (BAI), depression (BDI-II) and visual attention (TEA Map Search, DKEFS Trail Making Test) were also assessed. Results: Earlier VEPs (P1, N1), as well as processes related to visuospatial orientation (N2pc) and encoding in visual short-term memory (SPCN), appear comparable in mTBI and control participants. However, there appears to be a disruption in the spatiotemporal dynamics of attention (N2pc-Ptc, P2) in subacute mTBI, which recovers within six months. This is also reflected in altered neuropsychological performance (information processing speed, attentional shifting). Furthermore, orientation of attention (P3a) and working memory processes (P3b) are also affected and remain as such in the chronic post-mTBI period, in co-occurrence with persisting postconcussion symptomatology. Conclusions: This study adds original findings indicating that such a sensitive and rigorous ERP task implemented at diagnostic and follow-up levels could allow for the identification of subtle but complex brain activation and connectivity deficits that can occur following mTBI.

## 1. Introduction

In recent years, an increasing number of studies have shown the impact of mild traumatic brain injury (mTBI) on cognitive functions. The fact that mTBIs are considered a major health issue involving long-term health risks raises questions as to how their impact can be best identified and measured in order to be treated and managed optimally.

A TBI is produced by a large transfer of energy generated by a direct impact of the head against a hard surface, or by forces (acceleration, deceleration, rotation) created during the impact [1]. In the case of mTBI, these forces are responsible for stretching of the axons and microbleeds, which lead to a complex neurometabolic cascade [2]. There is now growing and reproducible evidence that following mTBI microstructural damage and neurochemical imbalances occur in a number of brain regions (e.g., frontal, temporal, motor cortex) and in white matter integrity (e.g., corpus callosum) [3,4,5,6,7,8]. These alterations have been related to the known post-mTBI physical (e.g., headaches, drowsiness, fatigue, dizziness), cognitive (e.g., attention and memory problems, bradyphrenia), and affective (e.g., irritability, depression, anxiety) symptoms [1,9,10]. These symptoms are more intense in the first days and weeks following the injury and slowly decrease during the subacute recovery period, which has been described as the first three months post-trauma. However, in a non-negligible proportion of individuals these symptoms are at risk of becoming chronic [11].

Attentional problems such as difficulties orienting and disengaging attention [12], sustaining it over time, and/or dividing attentional resources across multiple simultaneous demands [13,14] are amongst the most common complaints reported by individuals after a mTBI. On a behavioural level, these deficits usually disappear over a period of seven to 10 days [10,12,15]. However, mTBI can have negative long-term neuropsychological impacts on subtle aspects of complex attention and working memory [16,17], even with normal behavioural performances, where some individuals report the persistence of attentional difficulties that interfere with the demands of their daily life and social participation. The need for a more sensitive functional measure of the impact of mTBI on distinct steps involved in the deployment of attention thus remains.

Event-related brain potentials (ERPs) can represent a relatively simple, inexpensive, and precise answer to that question because they allow us to assess the integrity (reflected by the amplitude of a component) and efficiency (reflected by its latency) of specific and complex cognitive processes [1,18]. With excellent temporal resolution, ERPs possess an advantage over reaction time measures because they provide measures of multiple stages of cognitive processing, instead of the summation of the duration of all the intervening mechanisms involved in the generation of a response [18].

Experimental paradigms have historically been designed to study precise cognitive processes along with their event-related components. For instance, oddball paradigms have been used to evoke the P3, a component reflecting a target stimulus upgrade in working memory [19]. Such paradigms have been shown to be sensitive following mTBI [20,21,22,23,24]. However, since mTBI can present with deficits at different levels within the deployment of attention, one needs to be able to rapidly track the steps of the process that are distinctly affected.

The objective of this study was to track, with the help of a single task, the deployment of visuospatial attentional mechanisms and to identify underlying deficits at different recovery time points after mTBI. The task we used allowed us to measure the integrity of the different visual and cognitive processes involved in the deployment of visuospatial selective attention. We studied components reflecting earlier visual–attentional processing, such as the N1 (discrimination processes within the focus of visual attention and central attention) [25,26], P1 (facilitation effect for stimuli presented at an attended localisation) [27], and P2 (early attentional modulation) [28] components, which we expected would not be different between the mTBI and control groups. We also investigated later attention-related and visuocognitive mechanisms such as the N2pc (deployment and orientation of visuospatial attention) [28], Ptc (target isolation once identified among distractors) [29], SPCN (encoding in visual short-term memory, working memory capacity) [30], P3a (disengaging of attention to re-orient toward novel stimuli) [31], and P3b (tracking of task-relevant stimuli during updating in working memory) [19] components, which were expected to be affected, at least for later components.

## 2. Materials and Methods

### 2.1. Participants

Three groups of participants were enrolled in the study: 24 uninjured controls, 24 individuals in the subacute phase of mTBI (first three months post-mTBI), and 24 individuals in the chronic phase of mTBI (six months to one year post-trauma).

The uninjured control participants were recruited through publicity posted in a community centre. Exclusion criteria consisted of any neurological (history of brain trauma, seizures, attention deficit disorder, or learning disability) or psychiatric (depression, anxiety disorder, or other) antecedents. The mTBI participants had sustained their injury during an accident involving a motor vehicle or a fall. Diagnostic criteria for mTBI were: 1) Glasgow Coma Scale, on presentation at emergency room, between 13 and 15 (on a maximum total of 15); 2) having had any alteration in consciousness not lasting more 30 min or a post-traumatic amnesia duration of less than 24 h [32]. All participants with mTBI had been diagnosed by a rehabilitation medicine physician and were recruited from either a local trauma hospital or a rehabilitation centre. Exclusion criteria were the same as for the control participants. Additionally, participants with muscular–skeletal lesions or whiplash injuries sustained during or prior to the accident that caused the mTBI were excluded. The chronic mTBI group included eight participants from the subacute group who were re-tested. Also, according to event-related potential guidelines [18], all participants had to be at least 48 h removed from any alcohol or drug intake, and those under psychoactive medication were not included in the study. All participants had normal or corrected-to-normal vision.

Data from some participants had to be excluded from the analysis because of loss of trials (more than 50% of trials) due to eye blinks, movements, excessive sweating, or very large alpha oscillations that were mainly related to temporary temperature control issues in the testing environment. Analyses were performed on data from 22 participants in the control group (12 males, mean age 26.8 years, SD 6.6), 19 in the subacute mTBI group (17 males, mean age 36.6 years, SD 13.5, mean postinjury time 57 days, SD 19), and 17 in the chronic mTBI group (11 males, mean age 39.2 years, SD 13.5, mean postinjury time 271 days, SD 87). Groups were not quite equivalent for age (*F*(2,55) = 3.29, *p* = 0.045), with the control group being slightly younger, and there were more males (*X*^2^(2, *n* = 58) = 6.02, *p* = 0.049).

### 2.2. Procedure

A consent form, previously approved by the institutional ethics committee, was signed by the participants before their participation in the study. Neuropsychological and ERP testing lasted approximately 90 min, including electrode preparation/removal and frequent breaks. Participants received a financial compensation of 80 Canadian dollars for their participation.

#### 2.2.1. Neuropsychological Testing

To quantify the presence of self-reported symptoms, participants rated the Postconcussion Symptoms scale (PCS; 22 items, rated 0—asymptomatic to 6—severely symptomatic, for a maximum score of 132) [33]. Participants also completed the Beck Anxiety Inventory Scale (BAI; 21 items, rated 0—symptoms absent to 3—severe symptoms, for a maximum score of 63) [34] and the Beck Depression Inventory Scale–II (BDI-II; 21 items, rated 0—symptoms absent to 3—severe symptoms, for a maximum score of 63) [35]. Visual attention was tested with the Map Search subtest (number of target symbols circled among distractors within 1 and 2 min, maximum 80) of the Test of Everyday Attention (TEA) [36], and by the Delis-Kaplan Executive Function System (DKEFS) Trail Making Test (time to completion for conditions 1 to 5: Visual Scanning, Number Sequencing, Letter Sequencing, Number–Letter Sequencing, Motor Speed) [37].

#### 2.2.2. ERP Paradigm

The paradigm used in this study, which is presented in Figure 1, was first described by Bolduc-Teasdale, Jolicoeur & McKerral [38], where the detailed procedure, which was based on previously published paradigms [19,39,40], can be found.

#### 2.2.3. EEG recording and Analysis

We used a widely accepted EEG recording and analysis method, with a 64 Ag/AgCL scalp-electrodes montage, along with VEOG and HEOG electrodes and mastoid reference [38]. Number of trials included in EEG averaging were similar across conditions and groups, with a minimum of 35 trials per task condition.

Epochs were baseline-corrected based on mean amplitude of activity recorded over a period of 200 ms prestimulus. Subtraction methods were used to isolate specific ERP components. The P3a wave was obtained by averaging waveforms associated with irrelevant infrequent trials and frequent standard trials separately, and then subtracting the frequent standard stimuli activity from the irrelevant infrequent target signal. The latency of the P3a on the Fz electrode site was calculated by measuring the most positive point recorded between 350–540 ms for all groups [41]. The mean amplitude of this component was also quantified over the same time frame. To isolate the P3b component, infrequent response trials and frequent response trials were averaged separately. Activity for frequent stimuli was subtracted from the averaged infrequent target signal. The time window that was then used to quantify the P3b mean amplitude was between 500–615 ms for all groups. The latency of this component was calculated by taking the most positive point recorded within this predefined time window on the Pz electrode site [42].

The N2pc and SPCN components were obtained by separately averaging trials with either right or left visual targets over 1000 ms epochs, including a 200 ms prestimulus baseline. These components were obtained by subtracting ipsilateral neural activity (recorded over the hemisphere on the same side as the stimulated visual field) from contralateral neural activity (recorded over the hemisphere on the opposing side of the stimulated visual field). For all groups, the mean amplitudes were computed in time windows between 245 and 265 ms for the N2pc, and 380–680 ms for the SPCN. Previous work has shown that such lateralized components reached maximum peak amplitudes at the P07 and P08 electrode sites [28,43].

Using the same methodology as that previously described to obtain the N2pc and the SPCN (subtracting ipsilateral neural activity from contralateral neural activity), we also obtained the Ptc (positivity toward temporal electrodes contralateral). The analyses were conducted on the P07-P08 electrodes and the time window analysis was defined between 290–320 ms for all groups. The N2pc-Ptc peak-to-peak amplitude was calculated as a measure of spatiotemporal attentional efficiency.

P1, N1, and P2 waveform analyses were conducted on the Oz electrode site, between 80–90 ms, 140–170 ms, and 215–225 ms, respectively, for all groups.

For each component, maximal mean amplitude was measured in the indicated time window for each peak based on the grand average waveform. The latency was estimated using a semi-automatic peak detection function in the Brain Vision Analyzer program (Brain Products GmbH, Gilching, Germany). The indicated time windows set around the peak of the components and visual inspection assured that the peak detected corresponded with the maximal point of the component.

### 2.3. Statistical Analysis

Descriptive statistics were used to analyse the latency and amplitude data of each ERP component, behavioural data from the ERP recordings task, and neuropsychological data. Analysis of variance (ANOVA) was subsequently used for hypothesis testing across the three groups.

## 3. Results

### 3.1. Neuropsychological Results

Mean scores were computed for each measure, for the three groups, and are shown in Table 1.

Scores on the PCS for number of symptoms reported and total symptom score were submitted to a between-groups ANOVA. Results showed significant differences across groups for both number of reported symptoms (*F*(2,49) = 8.40, *p* = 0.001, ɳ^2^ = 0.23) and total symptom score (*F*(2,49) = 6.61, *p* = 0.003, ɳ^2^ = 0.19). Post hoc analysis showed that subacute and chronic mTBI groups reported more symptoms than controls (number of symptoms: *p* < 0.05; total score: *p* < 0.05). There was no significant difference between subacute and chronic mTBI (number of symptoms and total score: *p* > 0.05).

Analyses for the BAI and BDI-II were conducted on the total score for each group. There were no significant group differences for anxiety scores on the BAI (*F*(2,55) = 1.41, *p* = 0.25). There was a significant difference for depression scores on the BDI-II (*F*(2,55) = 4.10, *p* = 0.02, ɳ^2^ = 0.13), with post hoc analysis showing that the subacute mTBI group reported higher levels of depressive symptoms than the control group (*p* < 0.03), although not reaching the clinical criteria for depression (Beck, Sterr, and Garbin, 1988).

TEA Map Search analyses were conducted on the 1 min and 2 min conditions. There were no significant between-group differences for these two conditions (1 min: *F*(2,55) = 1.55, *p* = 0.22), 2 min: *F*(2,55) = 1.04, *p* = 0.36).

The D-KEFS Trails (conditions 1 to 5) analyses showed significant between group differences for Trail 2 (number sequencing) (*F*(2,55) = 3.17, *p* = 0.05, ɳ^2^ = 0.10), Trail 3 (letter sequencing) (*F*(2,55) = 4.14, *p* = 0.02, ɳ^2^ = 0.13) and Trail 4 (number-letter sequencing, switching condition) (*F*(2,55) = 3.62, *p* = 0.03, ɳ^2^ = 0.12). There were no between-group differences for Trail 1 (detection condition) (*F* (2,55) = 1.2, *p* > 0.31) or Trail 5 (motor control condition) (*F*(2,55) = 1.62, *p* = 0.21). Also, the three groups had equivalent number of errors (*F*(2,55) = 0.21, *p* = 0.82). Post hoc analyses showed that subacute mTBI participants had slower completion times than the control group for Trail 2 (*p* < 0.09), Trail 3 (*p* < 0.02) and Trail 4 (*p* < 0.03).

### 3.2. Task Performance

Mean percentage of accuracy was computed for each condition (same colour trials, infrequent position or colour, frequent position or colour, right hemifield, left hemifield), for the three groups. The first 400 trials were included in the analysis in order to eliminate possible practice and fatigue effects. Mean percentage of accuracy in the visuospatial attention task for each condition and for all groups are presented in Table 2. Results show that control as well as mTBI participants showed valid and reproducible behavioural data on the task. Groups were equivalent on all the conditions of the task (*F_s_* < 1).

Mean reaction times for correct responses were computed for each condition (same colour trials, infrequent position or colour, frequent position or colour, right hemifield, left hemifield), for each group. Results are presented in Table 3. For all individual conditions tested, there was a significant between group difference on mean reaction times (*F_s_*(2,55) ≥ 3.7, *p_s_* < 0.05), where subacute mTBI participants were slower than controls (*p* < 0.05).

### 3.3. Electrophysiological Results

Results showed no significant amplitude or latency differences between groups for P1 (*F*(2,55) = 1.737, *p* = 0.186; *F*(2,55) = 0.679, *p* = 0.52) and N1 components (*F*(2,55) = 0.232, *p* = 0.79; *F*(2,55) = 0.605, *p* = 0.55). Although it did not reach the significance level, there was a tendency toward a larger P2 amplitude in controls (*F*(2,55) = 2.783, *p* = 0.07). The were no significant latency differences for the P2 component (*F*(2,55) = 2.312, *p* = 0.11) (see Figure 2).

N2pc results show that there were no significant differences between groups for the amplitude of the N2pc (*F*(2,55) = 0.960, *p* = 0.39), nor for its latency (*F*(2,55) = 1.368, *p* = 0.26) (see Figure 3). The same was found for the amplitude of the Ptc component (*F*(2,55) = 1.07, *p* = 0.35), and its latency (*F*(2,55) = 2.616, *p* = 0.08). When comparing peak-to-peak amplitude shifts from the N2pc to the next-positive peak, the Ptc, the result was significant (*F*(2,55) = 4.25, *p* = 0.02, ɳ^2^ = 0.13), with a smaller amplitude shift between these components for subacute mTBI participants compared to controls (*p* < 0.05). Figure 3 also depicts averaged SPCN waveforms obtained on pooled electrodes PO7-PO8. There were no significant between-group differences for SPCN amplitude (*F*(2,55) = 0.7, *p* = 0.5).

There were significant differences between groups for the amplitude of the P3a (*F*(2,55) = 5.571, *p* = 0.01, ɳ^2^ = 0.17). Post hoc analysis showed that P3a was significantly larger for mTBI groups (subacute and chronic compared to controls: *p* < 0.05). There were no significant differences between groups for P3a latency (*F*(2,55) = 1.483, *p* = 0.24) (see Figure 4).

P3b results are shown in Figure 5. P3b amplitude was significantly smaller in both groups of mTBI participants (*p* < 0.05) relative to control participants (*F*(2,55) = 7.214, *p* < 0.002, ɳ^2^ = 0.21). The latency of the P3b was also significantly delayed in the mTBI groups (*p* < 0.05) compared to controls (*F*(2,55) = 7.839, *p* < 0.001, ɳ^2^ = 0.22).

## 4. Discussion

The electrophysiological paradigm implemented in the present study allowed us to measure several visuoperceptual and cognitive functions in a single session, resulting in a detailed examination and in original findings regarding the possible functional consequences of mTBI.

First, this study agrees with previous results by showing that, after a mTBI, earlier visual potentials are comparable to data from uninjured control participants [22]. It is, however, worth mentioning that there was a strong tendency for P2 amplitude reduction in mTBI compared to controls. This could suggest early attentional modulation difficulties in the form of less efficient visual search following mTBI [44].

Neuropsychological testing is recognized as an important part of the evaluation and follow-up of possible sequalae caused by mTBI [45]. It is now well known that, while symptoms sometimes recover within the few days after an injury, cognitive deficits may persist for longer [46]. The results obtained in this study are in accordance with the literature demonstrating that neuropsychological measures can be sensitive after a mTBI, at least in the subacute phase. Participants tested in the first three months post-mTBI showed slower processing speed in a visual attention task. While their basic selective visual attention abilities (TEA Map search, DKEFS Trail 1) appeared intact, they were negatively affected on a condition requiring rapid visuospatial processing (DKEFS: Trails 2 and 3), as well as in a condition known to target attentional and cognitive flexibility (DKEFS: Trail 4).

The chronic mTBI participants in this study did not show significant neuropsychological deficits. This result is in accordance with the literature, which often failed to demonstrate long-term effects on clinical neuropsychological testing after a single mTBI [47]. However, studies using more sensitive neurocognitive tasks measuring information processing speed and working memory have been able to show persisting cognitive impacts following mTBI [17,48]. Also, some studies have shown some neuropsychological deficits after two or more mTBIs [49,50].

In this study, the electrophysiological paradigm used revealed several statistically significant impacts, with large effect sizes, on neurocognition following a single mTBI. Previous work has found decreased amplitudes and increased latencies of the P3b component after mTBI [20,22,49,51]. Our results replicate such a pattern, where participants who sustained a mTBI showed significantly decreased amplitudes and increased latencies on the P3b component. Animal models have suggested that these impacts could be explained by the fragility of the hippocampic cells and of their related brain circuits, which are involved in updating the stimulus representation within working memory during an oddball paradigm and are thus reflected in the P3b component [52].

Neurometabolic and microstructural changes taking place after a mTBI were thus evidenced using the present ERP paradigm and can explain the decrease in efficiency and speed of associated cognitive processes. Indeed, participants who sustained a mTBI and were tested during the subacute phase (less than three months post-trauma) did have slower response times than uninjured controls and individuals with chronic mTBI during the ERP task, as well as on the neuropsychological tests requiring deployment of visuospatial attentional and executive processes. Another indicator that these cognitive processes were impacted by mTBI and were not as efficient as those of uninjured control participants is the fact that, while neuropsychological measures in the chronic mTBI group were comparable to those of the control group, P3b results showed persisting amplitude attenuation, even six months postinjury. This finding is in accordance with studies reporting the longer-term impact of mTBI on the P3b [21]. While the study of de Beaumont and al. [21] showed this type of result after multiple concussions, the present study demonstrates that such impairments can occur after a single mTBI.

Based on a study published by Halterman et al. [12], we assumed that orienting visuospatial attention would be affected in the subacute phase after mTBI. Indeed, they showed that individuals with mTBI were slower than controls in an orientation task, especially on the executive components of the task. In our ERP task, the N2pc component reflects these mechanisms, more precisely the attentional filter that allows the orientation of attention toward the target. Results obtained in this study did not show a significant impact of mTBI on the N2pc, although, as shown in Figure 3, visual inspection of the component appears to indicate an amplitude reduction in mTBI compared to controls at a more acute stage. This result is consistent with a previous study by de Beaumont et al. [49], who found no significant effect on that component at nine months post-trauma in athletes who sustained multiple concussions. The N2pc is a component of relatively small amplitude (often less than 3 µV) obtained from a subtraction (contralateral minus ipsilateral), and thus the difference score has the combined variance of the contralateral and ipsilateral waves. It is possible that more participants could have allowed a significant difference to emerge. Indeed, even a small decrease in the amplitude of the N2pc could be clinically significant without being statistically significant. The same pattern of results was obtained with the SPCN, which represents the coding of visual information in visual short-term memory. As the N2pc, the SPCN is of very small amplitude, so at this point, it is not possible to exclude the possibility that this component could be affected by a mTBI in larger groups of participants.

Further observation of the results allows the identification of the latency shift between the N2pc and the following positivity Ptc peak, which significantly differs between the control and subacute mTBI groups. The Ptc component is thought to reflect the process used to isolate the target once it is identified among distractors [29]. The shorter N2pc-Ptc peak-to-peak amplitude shift in the subacute mTBI group is another aspect that points to a disruption in the spatiotemporal dynamics of attentional processes following mTBI. This mechanism appears to be affected in the subacute phase but seems to recover among chronic mTBI participants. Such a fluctuation in these spatiotemporal processes could have important implications, especially when clinicians contribute to return to play decisions for athletes. Indeed, contact sports rely heavily on visual–attentional abilities and require one to be alert and respond rapidly in order to avoid re-injury. Knowing that another mTBI within a short time frame could ultimately result in a second impact syndrome, causing massive brain swelling and sometimes death [53], returning an athlete to play before the recovery of such neurophysiological alterations could have major impacts on the health of the player.

While these mechanisms appear to recover within a few months after mTBI, the results of the P3b differ in the way that the amplitude decrease and latency increase shown for this component, as well as postconcussion symptoms, remain significantly present in the chronic stage of mTBI. The results obtained for the P3a also show a long-term impact on the orientation of attention. Indeed, it appears that the mTBI participants (both subacute and chronic) had difficulties in disengaging their attention (as reflected in a much larger and sustained amplitude) once it had been directed toward the target. Such a result was also obtained for the subacute mTBI group on the neuropsychological task tapping into this process (D-KEFS: trail 4). Indeed, it measures cognitive flexibility, a process that allows one to disengage attention to be able to switch between different stimuli. While this task was affected in mTBI in the subacute phase, P3a results show that this disengaging process was still impaired after more than six months postinjury. These original results, which are along the same line as others obtained with different methodologies [13,14,17], underscore the importance of carefully monitoring, with appropriate tools, individuals with mTBI who need to go back to activities that require maintaining high attentional levels or multitasking (e.g., operation of heavy machinery, contact sports, driving).

The main limitation of this study is the small number of participants in each group and the cross-sectional design. Indeed, we were confronted with different recruitment and participation retention issues because of the nature of this clinical population. Also, some of the participants had to be removed from the analyses because they showed too many artefacts on their EEG. Performance accuracy rates were, however, very high in all groups. Among the participants included in the analyses, we noticed a relatively high degree of inter- and intra-individual variability. Nonetheless, effect sizes for significant results were large.

Future studies with a greater number of subjects will help to determine the optimal conditions for clinical use of this paradigm. This is especially relevant as this new task allowed us to track the deployment of visuospatial attention as reflected in several ERP components, ultimately providing a rich possible set of biomarkers for mTBI. For example, in some individuals with mTBI, we were able to observe a clear decrease of the N2pc, with a normal P3b and SPCN. For other participants, the P3b was decreased, while the N2pc and the SPCN were normal. These specific patterns were lost in the averaging process required for ERP analyses. It appears that the clinical interest of this ERP task lies in the gathering of normative data that could bring to light these different individual clinical neurocognitive patterns of visual–attentional processing after a mTBI.

## 5. Conclusions

It thus appears that such a complete, sensitive and rigorous task [38] implemented at the diagnostic level can provide a clear and specific window into brain functions and connectivity, allowing the identification of multiple and complex alterations in attention following mTBI. This highlights the necessity of measuring various aspects of information processing before drawing an early conclusion of complete recovery on the basis of a single variable or of postconcussion symptomatology alone.

## Figures and Tables

**Figure 1 brainsci-09-00343-f001:**
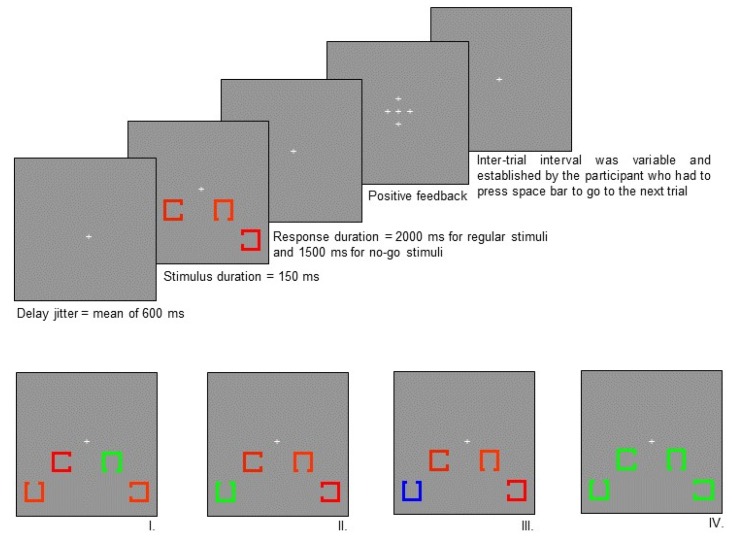
Experimental paradigm. *Top panel*: Experimental design. Each of the four squares subtended a visual angle of 1° × 1° with an opening of 0.33° on one side. Two squares were presented on each side of the fixation point. The centre of the squares nearest to the fixation point was 1.5° below and 3.5° to the left or the right of the fixation point. The centre of the farthest squares was 3° below and 5° to the left or right of the fixation point. The target square was presented equally often at each of the four possible positions (near left of fixation, near-right of fixation, far-left of fixation, far-right of fixation). The squares were in one of three different colours; blue, red, or green. The ERPs were evoked by manipulating the position of the opening of target squares, their colour, and the frequency of their occurrence. All these parameters were counterbalanced amongst participants such that the specific colours were not confounded with the various conditions in the experiment (the colours illustrated in Figure 1 represent the colour assignments for one of the many counterbalanced conditions). The intensity of the different colours of squares was calibrated to be equiluminant with a chroma meter (Minolta CS100) in order to control for low-level sensory responses. *Lower panel*: Stimuli. I. Frequent position of the square opening (standard stimulus), II. Infrequent position of the square opening (target stimulus), III. Infrequent colour of the target stimulus, IV. Same colour stimulus. Colour of different type of stimuli was counterbalanced among participants.

**Figure 2 brainsci-09-00343-f002:**
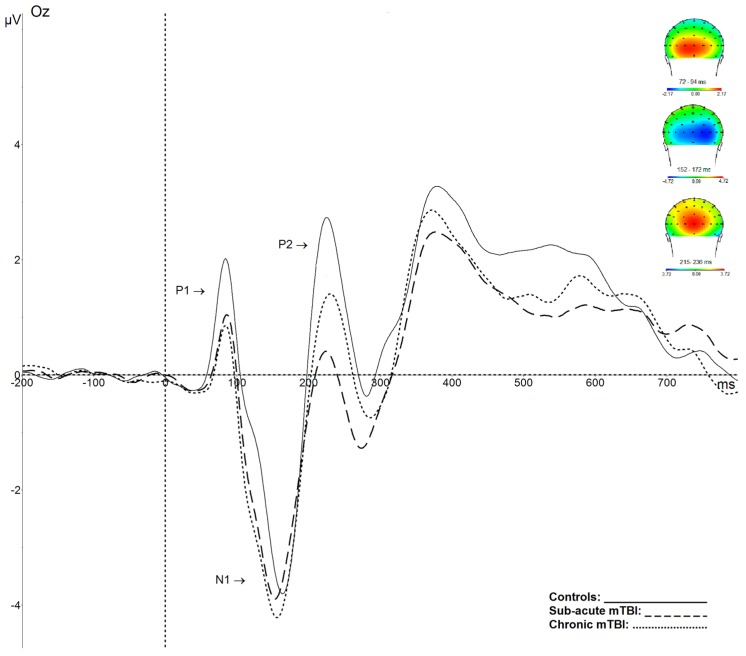
Grand average waveforms at Oz showing the visual P1, N1, and P2 components, and corresponding topographical maps, for controls (plain line), subacute mTBI (dashed line), and chronic mTBI (dotted line) participants. There is a tendency towards a larger P2 amplitude in controls.

**Figure 3 brainsci-09-00343-f003:**
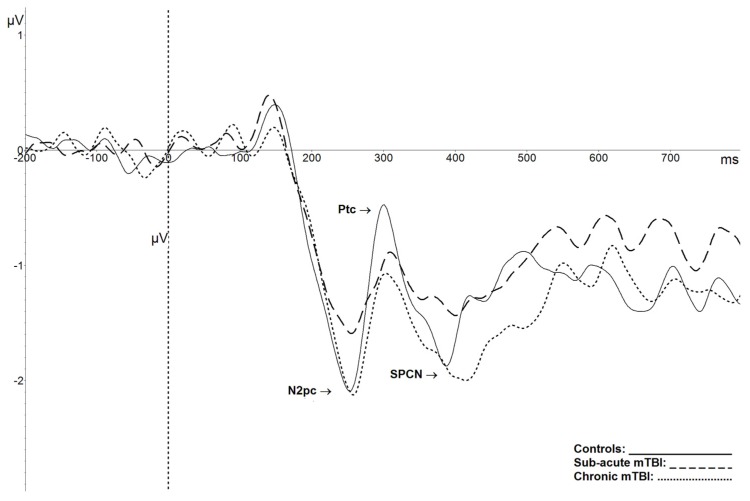
Grand average N2pc, Ptc, and SPCN components, evoked by lateralized stimuli, recorded at P07-P08 for controls (plain line), subacute mTBI (dashed line), and chronic mTBI (dotted line) groups. The N2pc-Ptc peak-to-peak amplitude shift is significantly smaller in the subacute mTBI group than in controls.

**Figure 4 brainsci-09-00343-f004:**
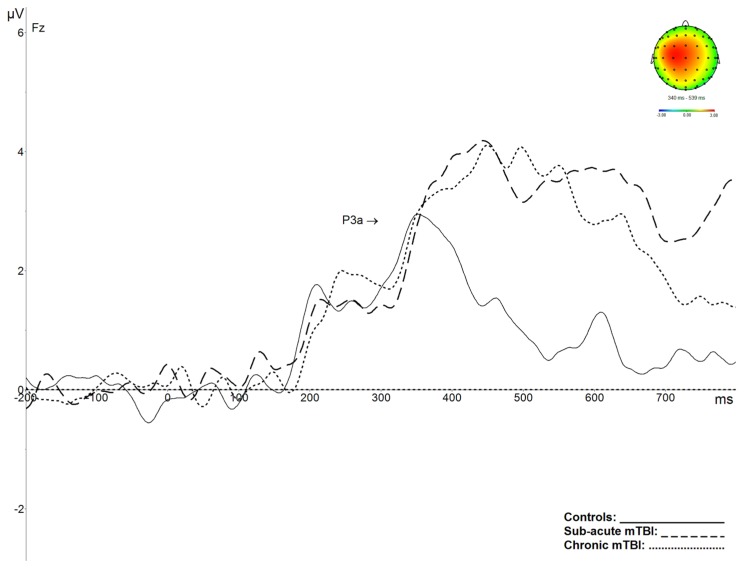
Average P3a components, obtained after the subtraction of activity evoked by same colour target and frequent colour standard stimuli, recorded at Fz, and corresponding topographical maps. The amplitude of the P3a component is significantly enhanced for subacute mTBI (dashed line) and chronic mTBI (dotted line) participants, in comparison with controls (plain line).

**Figure 5 brainsci-09-00343-f005:**
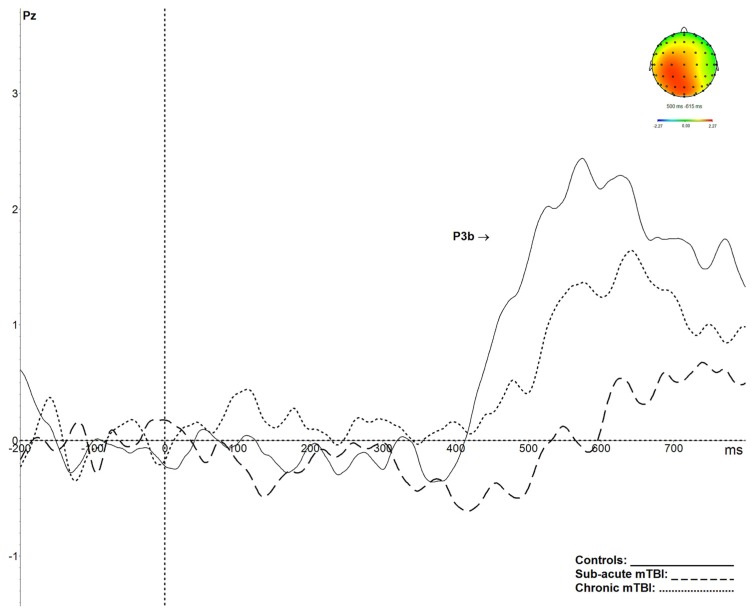
Grand average P3b components, obtained after the subtraction of activity evoked by infrequent position target stimuli and frequent position standard stimuli, recorded at Pz, and corresponding topographical maps. The P3b is significantly reduced and delayed in the subacute mTBI group (dashed line) and reduced in the chronic mTBI group (dotted line) in comparison with controls (plain line).

**Table 1 brainsci-09-00343-t001:** Neuropsychological test results for the three groups.

Task	Controls Mean (SD)	Subacute mTBI Mean (SD)	Chronic mTBI Mean (SD)	F
PCS Number of symptoms	2.97 (2.78)	8.69 (5.75)	7.37 (5.37)	8.78 *
PCS Total score	5.16 (5.76)	21.62 (18.12)	19.67 (21.04)	6.57 *
BAI	4.57 (3.72)	8.04 (7.60)	7.94 (11.69)	0.25
BDI-II	5.36 (3.97)	11.94 (8.23)	10.41 (10.43)	4.09 *
Map Search 1 min (targets)	59.64 (13.91)	52.58 (12.28)	53.90 (14.93)	1.55
Map Search 2 min (targets)	77.06 (4.85)	74.84 (6.88)	75.60 (7.38)	1.04
Trail 1 (s)	16.59 (2.92)	19.42 (8.20)	18.35 (5.82)	1.19
Trail 2 (s)	27.37 (7.63)	33.94 (9.52)	34.47 (12.93)	3.17 *
Trail 3 (s)	25.94 (7.28)	33.57 (9.89)	30.88 (8.73)	4.14 *
Trail 4 (s)	57.95 (15.70)	75.50 (29.28)	68.71 (15.52)	3.62 *
Trail 5 (s)	23.82 (10.23)	35.79 (32.09)	31.65 (17.84)	1.62

* Significant at *p* < 0.05.

**Table 2 brainsci-09-00343-t002:** Accuracy (percent) for each task condition, for the three groups.

Task Condition	Controls Mean (SD)	Subacute mTBI Mean (SD)	Chronic mTBI Mean (SD)
Same colour	96.3 (4.8)	95.7 (9.7)	92.8 (23.6)
Infrequent target position	90.3 (6.0)	83.1 (19.9)	90.2 (9.2)
Frequent target position	96.3 (4.3)	94.6 (4.3)	95.4 (5.7)
Infrequent target colour	95.8 (3.5)	89.7 (9.6)	90.3 (13.6)
Frequent target colour	94.6 (5.1)	92.3 (6.5)	95.1 (4.9)
Right	95.1 (3.7)	91.7 (7.4)	93.3 (7.2)
Left	94.5 (4.8)	91.8 (7.1)	94.9 (4.4)

**Table 3 brainsci-09-00343-t003:** Reaction time (ms) for each task condition, for correct trials, for the three groups.

Task Condition	Controls Mean (SD)	Subacute mTBI Mean (SD)	Chronic mTBI Mean (SD)
Same colour	No-go trial	No-go trial	No-go trial
Infrequent target position	769 (122)	876 (150)	801 (99)
Frequent target position	754 (129)	832 (143)	776 (138)
Infrequent target colour	778 (114)	878 (164)	830 (155)
Frequent target colour	752 (132)	832 (141)	771 (115)
Right	746 (123)	839 (144)	782 (137)
Left	769 (131)	842 (141)	781 (110)

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
