# Peer review of "Electrophysiological Markers of Visuospatial Attention Recovery after Mild Traumatic Brain Injury"

_brainsci, 2019, doi:10.3390/brainsci9120343_

Round 1

Reviewer 1 Report

In the current manuscript, the authors describe a study in which 3 populations (controls, those in the sub-acute phase, and those in the chronic phase post mTBI) are assessed behaviorally and electrophysiologically on attentional function. EEG components examined included the P1, N1, P2, N2pc, Ptc, SPCN, P3a and P3b, as well as differences between the N2pc and Ptc components. Results indicated that there were differences between controls and mTBI participants that were detectable via EEG, but not through behavioral testing. The authors conclude by emphasizing the importance of sensitive/complete assessment of the consequences of mTBI to ensure the safety of these individuals by determining when they have recovered sufficiently to return to potentially hazardous activities.

The methodology described is thorough, sound, and well justified, and the results are of great potential importance with real world application. There are a few issues with clarity throughout the manuscript that if addressed, may improve the readability and impact of the paper.

Specific Comments/Concerns:

The figures in the manuscript appear blurred/possibly low resolution and the font size is quite small in places. A higher resolution and fonts of at least 8 point would improve the readability of the figures.

Page 3

To give a better indication of the amount of data actually used in the analyses, could the authors also indicate the average percentage of trials removed due to artifacts for the remaining participants (after those with 50% or more trials rejected were removed)?

Page 5

The following sentence is a bit difficult to understand as it is currently phrased.

“Using the same subtraction that the one previously described to obtain the N2pc and the SPCN, we also obtained the Ptc (Positivity toward temporal electrodes contralateral).”

Starting on page 6

Could the authors include effect size estimates for each significant result?

Page 6

In the following sentence, it appears that the word “differences” may have been used one too many times.

“The D-KEFS Trails (conditions 1 to 5) analyses showed significant differences between group differences for Trail 2 (number sequencing) (F(2, 55)= 3.17, p = .05), Trail 3 (letter sequencing) (F(2, 55) = 4.14, p = .02) and Trail 4 (number-letter sequencing, switching condition) (F(2, 55) = 3.62, p = .03).”

Page 7-8

It appears that all analyses on the behavioral data made comparisons between groups, and within condition. Were there no predicted differences between conditions (e.g. performance on rare vs frequent target-position trials) or for interactions between group type and condition differences?

Page 8 Last Sentence

Is the statistic referring specifically to a lack of significant latency differences in P2? If so, it is a little confusing to preface this statistic by saying there were no significant differences in “P1, N1, and P2” since the data demonstrating this for P1 and N1 have already been related in previous sentences.

Page 8 Figure 2

While the correspondence between the different lines and groups is described in the figure caption, a legend on the figure itself would allow for easier/faster interpretation of this and other figures in the manuscript.

Page 9

While perhaps not of direct theoretical importance for the current study, there is no discussion of differences in the latency of the Ptc component between the three groups. For balance/completeness in the manuscript, it may be worth including this analysis so that amplitude and latency analyses are included for each component.

Page 9

Similarly, there does not appear to be a latency analysis for the SPCN.

Page 11 Last two paragraphs

It feels as if the same point about the P3b is discussed multiple times and could be said more efficiently/succinctly.

Pages 9 and 12

On page 9, data are presented describing amplitude shifts between N2pc and Ptc and it is stated that there is a greater amplitude shift for controls. Then on page 12, there is discussion of a significant latency shift difference for the N2pc to Ptc, with a “…longer N2pc-Ptc peak latency shift in the sub-acute mTBI group…” Are these two describing the same thing? If so, the language should be made consistent between the two. If these are not the same (e.g. one is describing differences in amplitude and the other is describing differences in latency), then there should also be latency analyses in the results section for the N2pc-Ptc comparison.

Reviewer 2 Report

A well-written informative paper which I believe adds some interesting and important data for those of us involved in the treatment of the mTBI population.

As I have commented, i agree that perhaps your results might have been more conclusive with larger groups - however could you have used a statistical package where participants' data could have been analysed individually. The TBI population is so lacking in homogeneity.
